# Development of Certified Reference Materials for the Determination of Apparent Amylose Content in Rice

**DOI:** 10.3390/molecules27144647

**Published:** 2022-07-21

**Authors:** Yafang Shao, Dawei Zhu, Jing Yu, Changyun Fang, Renxiang Mou, Xianqiao Hu, Zhiwei Zhu, Mingxue Chen

**Affiliations:** 1China National Rice Research Institute, Hangzhou 310006, China; nxyzdw@163.com (D.Z.); yjingyx@163.com (J.Y.); fcyst88@163.com (C.F.); mourx@hz.cn (R.M.); hxhxqia@aliyun.com (X.H.); zwzhu80@126.com (Z.Z.); 2Laboratory of Quality and Safety Risk Assessment for Rice (Hangzhou), Ministry of Agriculture and Rural Affairs, Hangzhou 310006, China

**Keywords:** rice, certified reference materials, apparent amylose content, calibration, spectrophotometry, starch

## Abstract

Apparent amylose content (AAC) is one of the most important parameters in rice quality evaluation. In this study, four rice reference materials used to test rice AAC were developed. The AAC of rice reference materials were measured by a spectrophotometric method with a defatting procedure, calibrated from potato amylose and waxy rice amylopectin at the absorption wavelengths of 620 and 720 nm. Homogeneity test (*n* = 20) was judged by F-test based on the mean squares of among and within bottles, and short- and long-term stability monitoring was performed by T-test to check if there was significant degradation at the delivery temperature of under 40 °C (14 days) and at 0–4 °C storage condition (18 months), respectively. After joint evaluation by ten laboratories, Dixion and Cochran statistical analyses were presented. The expanded uncertainties were calculated based on the uncertainty of homogeneity, short- and long-term stability, and inter-laboratory validation containing factor *k* = 2. It found that the four reference materials were homogenous and stable, and had the AAC (g/100 g, *k* = 2) of 2.96 ± 1.01, 10.68 ± 0.66, 17.18 ± 1.04, and 16.09 ± 1.29, respectively, at 620 nm, and 1.46 ± 0.49, 10.44 ± 0.56, 16.82 ± 0.75, and 24.33 ± 0.52, respectively, at 720 nm. It was indicated that 720 nm was more suitable for the determination of rice AAC with lower uncertainties. The determinations of the AAC of 11 rice varieties were carried out by two methods, the method without defatting and with calibration from the four rice reference materials and the method with a defatting procedure and calibrating from potato amylose and waxy rice amylopectin. It confirmed that the undefatted rice reference materials could achieve satisfactory results to test the rice samples with the AAC ranging from 1 to 25 g/100 g. It would greatly reduce the time cost and improve testing efficiency and applicability, and provide technical support for the high-quality development of the rice industry.

## 1. Introduction

Rice is one of the most important staple foods and provides an energy source for about one-third of the world’s population. With the improvement of people’s living standards, the demand for high-quality rice is increasing day by day. The quality of rice could reflect consumer acceptance directly and is recognized as the most important feature. Starch, accounting for about 90% of rice grains on a dry basis, consists of amylose and amylopectin, with the former being linear polymers and the latter being branched forms. Amylose is a mixture of a broad range of polymers with different molecular weights and configurations [1,2]. According to the degree of polymerization of amylopectin branches, it can be divided into chain A, long chain B, intermediate B, and short chain B. The apparent amylose content (AAC) comprising of the contents of amylose and the long chain B of amylopectin was considered to be the main factor influencing the texture of cooked rice. It was reported that rice with a higher AAC always had a harder and less sticky texture after cooking than that with a lower AAC [3,4]. It may be because that heating would destroy the amylose double helices resulting from the occurrence of two intracrystalline molecules of water and a tight network of hydrogen bonds involving each of the primary and secondary hydroxyl groups of the glucosyl moieties, and amylose molecules entangle or co-crystallize with amylopectin chains in the crystalline lamellae, which in turn limits starch swelling and leaching during cooking, thereby resulting in a harder and less sticky texture [5,6,7]. In addition, AAC could affect the pasting properties, gelatinization, texture, solubility [8,9], and the formation of resistant starch [10]. It was reported that the rice with high AAC had a lower glycemic index [11]. AAC could affect rice properties, and rice with different AAC could be processed into different kinds of rice products. For example, high AAC rice was preferred for processing noodles or bread, whereas waxy and/or low AAC rice was preferred for processing biscuits or crackers [9]. Even in parboiled processing, the expansion ratio of rice was dependent on its amylose content [12]. Therefore, the determination of AAC is an important factor in the selection of new rice varieties in breeding projects, and it also plays an important role in improving the quality of rice products.

Rice flour is suitable as a reference material because it is homogenous, stable, has low volatility, and has low toxicity, which ensures ease of management. In China, rice quality needs to be evaluated by regional tests, cultivar registration, and promotion, and AAC is one of the most important quality indexes. With the increasing demand for high-quality rice and the sharp increase in AAC test samples in recent years, it is very important to use rice AAC-certified reference materials to reduce the testing cost, ensure the quantity of quality control, and improve the accuracy and applicability of detection. Currently, there are many rice reference materials reported in the literature, such as rice elements reference materials, which solved the problem of detection accuracy caused by trace amounts [13], and rice pesticide reference materials, which conquered the challenge of the extraction of native compounds caused by solute–matrix interactions [14], rice genomic DNA reference materials, which had precisely defined quantity of template [15], and so son. However, the development of rice AAC-certified materials has been rarely reported. For rice AAC reference materials, it could reduce the time cost by removing the degreasing process in rice sample tests and reduce the matrix effect on low-amylose samples.

There are many methods to measure the AAC of rice grains, instrumental methods (e.g., near-infrared reflectance spectroscopy (NIR) [16] or NIR associated with suitable multivariate regression methods [17,18], the amperometry or potentiometric titration method [19], high-performance size exclusion chromatography [20], electrochemical method [21], the thermo-gravimetric method [22], the thermal method [23]), and the conventional iodine staining colorimetric method. The instrumental methods need expensive equipment, and some need tedious modeling steps and have poor sensitivity [16]. The conventional iodine staining colorimetry is the most widely used method. It is based on the principle that the double helix structure of the amylose can combine with the iodine to form a blue complex with a characteristic absorption peak at the wavelengths of 620 nm or 720 nm, and the AAC is calculated according to a calibration solution of potato amylose and waxy rice amylopectin [24,25]. In fact, the rice samples should be defatted before the AAC test, and the properties of raw starch would be affected during the long degreasing process. Moreover, the matrices of the prepared standard solution are close to but not exactly the same as all of the rice samples. Although the labor-intensive operations of the AAC test were simplified with the advent of the automatic continuous flow analyzer [26], the degreasing process of the rice samples was still the most important rate-limiting step. Therefore, how to break through the main speed-limit step is crucially important to improve the detection efficiency of rice AAC.

In this study, the AAC of four different conventional rice varieties was measured by a defatting procedure and calibrated by potato amylose and waxy rice amylopectin at two absorption wavelengths of 620 and 720 nm. After homogeneity, short- and long-term stability assessments, inter-laboratory validation by ten reputable national laboratories, and an expanded uncertainty (*k* = 2) calculation, they were developed as rice AAC reference materials. Eleven rice samples with AAC ranging from 1.0 to 25.0 g/100 g were selected and used to verify the accuracy and reproducibility of the reference materials by two methods, the method without defatting and with calibration from the four reference materials and the method with a defatting procedure and calibrating from potato amylose and waxy rice amylopectin. Meanwhile, the results of the two wavelengths were compared, and a better wavelength was obtained for the determination of rice AAC. The objective of this study was to establish a set of certified reference materials for rice AAC determination at two absorption wavelengths of 620 and 720 nm. It would greatly reduce the time cost, ensure the quantity of quality control, and improve the testing efficiency and applicability. Rice quality evaluation and cultivar registration would be accelerated greatly, which would be beneficial for the rice breeders to speed up the breeding of high-quality rice varieties.

## 2. Materials and Methods

### 2.1. Rice Samples

Four conventional rice grains, RM01 (Waxy rice L), RM02 (Jiayin 2), RM03 (Wujing 31), and RM04 (Zhongzao 39), were chosen as the objects for rice AAC reference materials. Their AAC ranged from 0 to 4%, 9 to 12%, 14 to 18% and 24 to 27%, respectively. They were planted in environments that were suitable for their growth. RM01 and RM04 were cultivated in the fields at Fuyang and Ruian in 2016, respectively. RM02 and RM03 were cultivated in the field at Danyang in 2016. In addition, 11 rice samples with the AAC ranging from 1.0 to 25.0 g/100 g were used to verify the accuracy and reproducibility of the four developed reference materials by two methods, the method without defatting and with calibration from them and the method with a defatting procedure and calibration by potato amylose and waxy rice amylopectin. All of the samples were sun-dried to a moisture content of less than 13% and stored under darkness at 4 °C. Prior to analysis, each grain was dehusked and polished on a mill (Yamamoto Co., Yamagata, Japan) and then passed through a 100-mesh sieve on a Cyclone sample mill (UDY Co., Fort Collins, CO, USA). Each sieved rice flour was tumbled on a V-type mixer (Changzhou, China) for 4 h and then divided into 1000 bottles with 4 g each.

### 2.2. Chemicals

Analytical grade methanol and ethanol, sodium hydroxide, acetic acid, and potassium iodide were purchased from Sinopharm Chemical Reagent Co., Ltd. (Shanghai, China). The defatted potato amylose and rice amylopectin with a purity of 97.1% and 88.3%, respectively, were ordered from Quality and Safety of Agricultural Products, Heilongjiang Academy of Agricultural Sciences (Harbin, China). As shown in Appendix A, the potato amylose met the requirements of ISO 6647-1 [24] with an iodine-binding capacity of 19.28% and a maximum absorbance of the iodine–starch complex at 542 nm.

### 2.3. Determination of Moisture Content

The moisture content (MC) was measured according to ISO 712. In brief, a quantity of 3.0 ± 0.5 g of rice flour (nearest 0.0001 g) was placed into a dry dish, and the mass of the undried sample and the dried dish (with its lid) was recorded as m_0_, and the mass of the previously dried dish with its lid was recorded as m_d_ (nearest 0.0001 g). After the open dish containing rice flour and together with the lid was dried in an oven for 4 h under 105 ± 2 °C, the dish was quickly removed from the oven, covered, and placed in a desiccator. After allowing the dish to cool to the laboratory temperature (about 30–45 min after being placed in the desiccator), it was weighed to the nearest 0.0001 g, and the mass of the dried rice flour and dish was recorded as m_1._ The moisture content of rice flour was calculated as follows:(1)$$MC\;\left(\%\right)=\frac{m_{0}-m_{1}}{m_{0}-m_{d}}\times 100$$

### 2.4. Determination of Rice AAC

The AAC of the four reference materials was tested by the colorimetric method [24] with minor modification. Approximately 1.0 g of rice flour were defatted by 20 mL of methanol for 6 h using a Soxhlet drawer system HT1043 extraction unit (Tecator, Höganäs, Onsala, Sweden), and then spread in a tray and left for 2 days to allow for the evaporation of the residual methanol and for moisture content equilibrium. After weighing 50.0 ± 0.1 mg of the defatted rice flour into a 50 mL conical flask, 0.5 mL of 95% ethanol was added using a pipette to wash down any of the flour adhering to the side of the flask. After shaking slightly to wet the entire sample, 4.5 mL of 1 M NaOH was pipetted into the flask, and it was heated in a boiling water bath for 10 min to disperse the rice starch completely. The mixture was allowed to cool down to room temperature and transferred to a 50 mL volumetric flask. After making up the volume with distilled water, the mixture was mixed vigorously and used as a crude extract in the following test. The blank, potato amylose, and rice amylopectin stock solutions (1.0 mg/mL) were carried out by the same procedure and reagents, except that the weights of potato amylose (51.5 ± 0.1 mg) and waxy rice amylopectin (56.6 ± 0.1 mg) were converted into a standard concentration according to its purity. The AAC calibration solutions of 0, 10, 20, 25, 30, and 35% were prepared by adding 0, 2, 4, 5, 6, 7 mL of 1.0 mg/mL potato amylose stock solution and 18, 16, 14, 13, 12, and 11 mL of 1.0 mg/mL rice amylopectin stock solution into 2 mL of 0.09 mol/L NaOH, respectively. For the color development, 2.5 mL of sample extracts or amylose calibration solutions were added into 50 mL volumetric flasks containing about 25 mL of water. After pipetting 0.5 mL of 1 M acetic acid solution and 1.0 mL of iodine solution, the mixture was fully mixed, followed by making up to the mark with water. After incubating for 10 min at room temperature, the absorbances were measured at the wavelengths of 620 and 720 nm.

Eleven rice samples were used to test the AAC by the method without defatting and with calibration from the developed rice reference materials [25] and by the method with a defatting procedure and calibrating from potato amylose and waxy rice amylopectin [24]. The results were expressed as grams of amylose per 100 g of rice flour on a dry basis (g/100 g). Duplicate determinations were conducted per extract.

### 2.5. Homogeneity Examination

According to ISO Guide 35 [27], the randomly selected samples should reflect the overall characteristics of the samples. When the total number of samples is less than 1000, 15–20 units should be selected. When the total number of samples is equal to or more than 1000, the quantity of the homogeneity test should be not less than 2 × $\sqrt[{3}]{N}$, where *N* denotes the total number of the reference material. Therefore, 20 sets of the reference materials were randomly selected with a triplicate determination of AAC. The homogeneity test was performed with SAS version 8 software (SAS Institute, Cary, NC, USA). The among-bottle variance is obtained from the relationships between the mean squares among and within groups and the expectations of the corresponding variance [28,29]. The mean squares among the different samples (*MS*_among_) and within the replications of the same samples (*MS*_within_) were found using the analysis of variance (ANOVA).

If *MS*_among_ ≥ *MS*_within_, the homogeneity is considered good. The among-bottle standard deviation (*S*_b_) is calculated by the formula as follows:(2)$$S_{b}=\sqrt{\frac{{MS}_{\mathrm{among}}-{MS}_{\mathrm{within}}}{n}}$$
where *n* is the number of the replication in one bottle. In this method, the pooled repeatability standard deviation equals $\sqrt{MS_{\mathrm{within}}}$ [30].

If *MS*_among_ < *MS*_within_, the differences due to within-bottle homogeneity are large and its repeatability is insufficient to estimate such differences, or there are outliers in the database. Alternatively, the among-bottle standard deviation (*S*’_b_) can be estimated as:(3)$${S^{\prime }}_{b}=\sqrt{\frac{MS_{\mathrm{within}}}{n}}\sqrt[{4}]{\frac{2}{a\times \left(n-1\right)}}$$
where *a* and *n* denote the number of bottles and the number of the replication in one bottle, respectively [30,31]. The among-bottle standard deviation reflects the uncertainty associated with homogeneity [27].

### 2.6. Stability Monitoring

The long- and short-term stabilities were measured. For the investigation of the long-term stability, three sets of reference materials were randomly selected from the refrigerator (0–4 °C) with the storage periods of 0, 3, 6, 9, 12, and 18 months. In order to investigate possible degradation during delivery, the short-term stability was assessed [32]. In the short-term stability test, every three sets of rice reference materials were put under three different temperature conditions (0–4, 19–21, and 39–41 °C) and four storage periods (0, 3, 7, and 14 days). Each set was tested in triplicate.

The statistical analyses of all the stability investigations were carried out to study if the apparent amylose of the reference materials degraded significantly during storage [32]. The slope of the fitted regression function was tested for significance at a 95% level [31]. If the slope of the regression line did not differ significantly from zero, the standard error of the slope was multiplied to derive an estimate of the uncertainty due to the possible instability of the materials [32,33].

According to ISO Guide 35 [27], the standard deviation at each point on the regression line can be calculated as:(4)$$S=\sqrt{\frac{\sum_{i=1}^{n}{\left(Y_{i}-b_{0}-b_{1}X_{i}\right)}^{2}}{n-2}}$$
where *b*_1_ denotes the slope
(5)$$b_{1}=\frac{\sum_{i=1}^{n}\left(x_{i}-\bar{x}\right)\left(Y_{i}-\bar{Y}\right)}{\sum_{i=1}^{n}{\left(x_{i}-\bar{x}\right)}^{2}}$$
and *b*_0_ denotes intercept
(6)$$b_{0}=\bar{Y}-b_{1}\bar{X}$$

The standard deviation of the slope can be calculated according to the formula as follows:(7)$$s\left(b_{1}\right)=\frac{S}{\sqrt{\sum_{i=1}^{n}{\left(X_{i}-\bar{X}\right)}^{2}}}$$

The uncertainty of long- and short-term stability is calculated as:(8)$$u_{\mathrm{lts}}\;\mathrm{or}\;u_{\mathrm{sts}}=n\times s\left(b_{1}\right)$$
where *n* indicates the storage time.

### 2.7. Collaborative Validation Study

Ten laboratories were selected for the inter-laboratory validation study. All laboratories used the same method to test the rice AAC by using two wavelengths of 620 and 720 nm. All of the measurements were conducted in triplicate in all the laboratories. The raw data obtained from the ten laboratories were tested for normal distribution and checked by the Dixion consistency test and Cochran statistical analysis to discard the outliers, and then used to calculate the mean values and the standard uncertainties [31,34].

The uncertainty of the results from the 10 laboratories was calculated as:(9)$$u_{\bar{\mathrm{char}}}=\frac{s_{\bar{\bar{x}}}}{\sqrt{n}}$$
where *n* denotes the number of laboratories, and
(10)$$s_{\bar{\bar{x}}}=\sqrt{\frac{\sum_{i=1}^{n}{\left({\bar{x}}_{i}-\bar{\bar{x}}\right)}^{2}}{n\left(n-1\right)}}$$

As the potato amylose is not a pure substance, the total uncertainty of the characteristic value should be calculated as:(11)$$u_{\mathrm{char}}=\sqrt{u_{\bar{\mathrm{char}}}{}^{2}+u_{b}{}^{2}}$$
where *u*_b_ denotes the uncertainty of the potato amylose standard.

### 2.8. Calculation of Expanded Uncertainty

The expanded uncertainties are comprised of the uncertainty induced by inhomogeneity, instability (long- and short-term), and different laboratories. The expanded uncertainty of the reference material was calculated as:(12)$$U=k\;\sqrt{u_{\mathrm{bb}}^{2}+u_{\mathrm{lts}}^{2}+u_{\mathrm{sts}}^{2}+u_{\mathrm{char}}^{2}}$$
where *u*_bb_ is the uncertainty associated with homogeneity, *u*_lts_ and *u*_sts_ denote the uncertainty contribution from long- and short-term stability, respectively, and *u*_char_ refers to the uncertainty of inter-laboratory validation.

### 2.9. Statistical Analysis

The AAC of the 11 rice samples was presented on a dry matter basis as means ± standard deviation of triplicate determinations. The data analyses were performed with SAS version 8 software (SAS Institute, Cary, NC, USA). Differences in the same sample by two different test methods at 620 and 720 nm were found by using ANOVA, followed by Duncan multiple comparison tests. The statistical significance was defined at a level of *p* < 0.05.

## 3. Results and Discussion

### 3.1. Homogeneity Examination

The moisture contents of the rice reference materials are shown in Appendix A. It was tested for five replicates and ranged from 9.78 to 11.35%. The analysis of homogeneity of the four rice reference materials was analyzed using ANOVA and is shown in Table 1. The average AAC of RM01–RM04 detected at the wavelength of 620 nm was 2.98, 10.54, 16.90, and 26.00 g/100 g, respectively. The average AAC of RM02 and RM03 detected at the wavelength of 720 nm was 10.42 and 16.60 g/100 g, respectively, which were similar to that detected at 620 nm. The average AAC of RM01 and RM04 detected at the wavelength of 720 nm was 1.47 and 24.05 g/100 g, respectively, both of which were lower than that detected at 620 nm. For all the results, the mean squares among the different samples (*MS*_among_) were larger than that within the replications of the same samples (*MS*_within_). The F values were calculated by dividing *MS*_among_ by *MS*_within_. The estimated F values (1.37–1.78) were compared with the critical Fα value at the *p* = 0.05 level of significance with *n*_1_ = 19 and *n*_2_ = 40 from the F-distribution table (Fα = 1.85) to check the significance of the results. It suggested that all of the F values were less than the Fα value. Therefore, the four reference materials were considered to be homogeneous.The *u*_bb_ was calculated by Equation (2), and the values tested at 620 nm and 720 nm ranged from 0.032 to 0.085 and 0.026 to 0.065, respectively.

### 3.2. Stability Monitoring

The stability monitoring of the four reference materials was conducted in two forms, long- and short-term. The long-term stability was measured under 0–4 °C for 18 months. The short-term stability was measured under 0–4, 19–21, and 39–41 °C for 14 days.

The long- and short-term stability analyses are shown in Table 2 and Table 3, respectively. The linear relationship according to month or day (independent variable) and AAC (dependent variable) was built, and the slope of the line and its significance (at 95% confidence level) was tested. According to two-tail confidence level of 95% in T-distribution table, the value of *t*
_(0.95, *n*−2)_ was 2.667 and 4.303 for long-term (*n* = 6) and short-term (*n* = 4) stability monitoring, respectively. As the estimated slopes (*b*_1_) were less than *t _(_*_0.95, *n*__−__2)_ × *s* (*b*_1_), they were considered statistically insignificant. Furthermore, the stability of the reference materials stored at 0–4 °C would continue to be monitored to extend their expiry date. The short-term stability was monitored at three temperatures, and it indicated that the rice AAC reference materials should be transported below 40 °C in order to ensure their quality strictly. The uncertainty of long- and short-term stability of the four reference materials ranged from 0.040 to 0.356 and from 0.029 to 0.493, respectively.

### 3.3. Collaborative Validation and the Expand Uncertainty Assessment

The AAC of the four rice reference materials was collaboratively validated by 10 laboratories (Table 4). The Shapiro–Wilk test method was used to check the normal distribution of all the laboratory means (*n* = 10). The calculated W-values ranged from 0.86 to 0.96, which were greater than the critical values of 0.09–0.75 (*p* = 0.95). The means obtained by the ten laboratories were considered to be in a normal distribution. The results of the Dixion consistency test and Cochran statistical analysis showed that there were no outlying variances or outliers in this set of data at a confidence level of 95%. Therefore, the mean value of all the individual results could be assigned as the certified value of rice AAC reference materials. The average AAC (detected at 620 nm) of RM01-RM04 was 2.96, 10.68, 17.18, and 26.09 g/100 g, respectively, and the uncertainty was 0.104, 0.126, 0.144, and 0.198 g/100 g, respectively. For the wavelength of 720 nm, the average AAC of RM01–RM04 was 1.46, 10.44, 16.82, and 24.33 g/100 g, respectively, and the uncertainty was 0.064, 0.068, 0.128, and 0.145 g/100 g, respectively.

The expanded uncertainties were calculated according to Equation (12). According to previous studies [27,35], short-term stability was used to predict the rate of change at a range of temperatures, and whether the short-term stability must be considered in the expanded uncertainty assessment or not need to be further studied. The expand uncertainties (*k* = 2) of RM01-RM04 (detected at 620 nm) were 1.01, 0.66, 1.04, and 1.29 g/100 g, respectively, and they were 0.49, 0.56, 0.75, and 0.52 g/100 g, respectively, at the detection wavelength of 720 nm (Table 5). The uncertainties of the four reference materials at 720 nm were lower than that at 620 nm, which indicated that the absorption wavelength of 720 nm is more suitable for the determination of AAC in rice.

### 3.4. Application

Rice AAC ranged from 7.9 to 33 g/100 g in normal rice and from 0 to 2 g/100 g in waxy rice [9]. It could be divided into five groups such as waxy (0–2%), very low (5–12%), low (12–20%), intermediate (20–25%), and high (25–33%), and most of the rice grains had the AAC ranged from 10 to 25% [9,36]. With the cultivation of semi-waxy japonica rice varieties and the quality improvement of *indica*-hybrid rice, more than 90% of the rice varieties had an AAC of less than 25 g/100 g, and about 30% of the rice samples belonged to waxy (0–2%) and very low groups (5–12%) in China [37,38]. Although there were three rice amylose content reference materials (bcr 465–467) in Europe, their use in China had two drawbacks: (1) the measuring linear range was 15.40–27.71 g/100 g, which could not be used for the determination of AAC in more than 30% of rice varieties; (2) the absorption wavelength was set at 620 nm only, which could not be used at the wavelength of 720 nm. In this study, the four rice reference materials had the AAC ranging from 2.96 to 26.09 g/100 g and from 1.46 to 24.33 g/100 g at the detective wavelengths of 620 nm and 720 nm, respectively. According to China Rice Industry Development Report 2021, they covered about 90% of rice varieties. There was no certified AAC reference material in China, and it was first proposed and successfully developed and certified.

The AACs of 11 rice samples were tested by ISO 6647-1 [24] and ISO 6647-2 [25] simultaneously in order to test the accuracy and reproducibility of the four certified rice reference materials. As seen in Table 6, the AAC of 11 rice samples ranged from 0.5 to 25.0 g/100 g. The correlation coefficients (R) of the two standard curves detected at the wavelengths of 620 and 720 nm were ≥0.998. The AAC under the wavelength of 620 nm was higher than that under the wavelength of 720 nm except for the samples of S06–S09. Most rice samples had similar AAC when using the two different standards. The absolute differences between AS and RM of all the 11 rice samples were less than the repeatability limit *r*. It indicated that the AAC measured by using the four rice reference materials as standards without defatting was consistent with that measured by potato amylose and rice amylopectin standards with the defatting procedure. It indicated that the four undegreased certified rice reference materials could be directly used as standards in the determination of rice AAC at wavelengths of 620 nm and 720 nm.

Certified reference materials play an indispensable role in the calibration of measuring instruments, the evaluation of analysis methods, and the measurement of material characteristics, as well as in the production process of product quality control. As with other tissue-certified reference materials, the transformation and accumulation of many substances occurred in rice grains, which made the preparation process of rice reference materials complicated and time-consuming. On the other hand, the AAC of rice grains could be affected by the environment, especially the growth temperature during grain filling [39,40,41]. Therefore, an assessment of homogeneity, long-term and short-term stability, and certified value of rice AAC reference materials needs to be carried out again in the new round of reference materials development. The moisture content of the five replications for each rice reference material ranged from 9.77 to 11.34%, and all of the samples were stored at a temperature of 0–4 °C. It suggested that these samples tend to absorb moisture from the air after being removed from the refrigerator. Therefore, they need to be kept in a dryer until the temperature matches the ambient temperature before testing.

## 4. Conclusions

Four rice reference materials (RM01, RM02, RM03, and RM04) used to test rice AAC were developed. AAC of the four rice reference materials were measured by a spectrophotometric method with a defatting procedure and calibration from potato amylose and waxy rice amylopectin at the wavelengths of 620 and 720 nm. The State Administration for Market Regulation in China has approved and numbered these matrix-certified reference materials as GSB 11-3875-2021. The certified values (g/100 g, *k* = 2) were 2.96 ± 1.01, 10.68 ± 0.66, 17.18 ± 1.04, 26.09 ± 1.29 at the detection wavelength of 620 nm, respectively, and 1.46 ± 0.49, 10.44 ± 0.56, 16.82 ± 0.75, 24.33 ± 0.52 at 720 nm, respectively. The rice AAC reference materials showed inhomogeneity among and within bottles of less than 5%. The stability assessment indicated that the rice AAC reference materials were stable for at least 18 months at 0–4 °C storage conditions and 14 days at a delivery temperature under 40 °C. The stability would be monitored continuously, and the expanded uncertainties would also be comprehensively evaluated. The determination of AAC of 11 rice samples was carried out using two methods, one without defatting and with calibration from the four rice reference materials and the other using a defatting procedure with calibration from potato amylose and waxy rice amylopectin. It was confirmed that the undefatted rice reference materials could achieve satisfactory results. The development of the rice AAC reference materials would greatly reduce the time cost, ensure the quantity of quality control, and improve the efficiency and applicability of detection. It would be of great importance to accelerate the process of rice quality evaluation and cultivar registration and promotion, which would be beneficial for the rice breeders to speed up the breeding of high-quality rice varieties and promote the development of the rice industry.

## Figures and Tables

**Table 1 molecules-27-04647-t001:** The estimated uncertainties of the homogeneity of AAC of the four rice reference materials ^a^.

	620 nm				720 nm			
	RM01	RM02	RM03	RM04	RM01	RM02	RM03	RM04
AVG	2.98	10.54	16.9	26	1.47	10.42	16.6	24.05
*df* _among_	19	19	19	19	19	19	19	19
*df* _within_	40	40	40	40	40	40	40	40
*MS* _among_	0.008	0.045	0.049	0.045	0.008	0.046	0.036	0.039
*MS* _within_	0.005	0.026	0.028	0.027	0.006	0.033	0.024	0.027
*u* _bb_	0.032	0.079	0.085	0.078	0.026	0.065	0.064	0.061
SD	0.05	0.12	0.13	0.12	0.05	0.13	0.11	0.11
F	1.62	1.73	1.78	1.66	1.38	1.38	1.51	1.42
Fα	1.85	1.85	1.85	1.85	1.85	1.85	1.85	1.85

^a^ AAC of the four rice reference materials were detected at the wavelengths of 620 and 720 nm (randomly 20 bottles with triplicate detection), and the result are present as g/100 g on dry matter bases. AVG: average content; *df*_among_: degree of freedom among bottles; *df*_within_: degree of freedom within bottles; F: the calculated F value; Fα: F value at *p* = 0.05, *n*_1_ = 19, *n*_2_ = 40; *MS*_among_: mean square among bottles; *MS*_within_: mean square within bottles; SD: standard deviation among bottles; *u*_bb_: uncertainty of homogeneity.

**Table 2 molecules-27-04647-t002:** Analyses of long-term stability of AAC of the four rice reference materials (0–4 °C) ^a^.

Time (Months)	620 nm				720 nm			
	RM01	RM02	RM03	RM04	RM01	RM02	RM03	RM04
0	3.33 ± 0.09	10.74 ± 0.08	16.75 ± 0.13	26.12 ± 0.18	1.42 ± 0.06	10.51 ± 0.08	16.54 ± 0.15	24.12 ± 0.19
3	2.90 ± 0.07	10.74 ± 0.13	17.02 ± 0.12	26.05 ± 0.12	1.52 ± 0.05	10.68 ± 0.15	16.81 ± 0.11	24.06 ± 0.15
6	2.68 ± 0.02	10.39 ± 0.13	17.26 ± 0.09	25.58 ± 0.09	1.49 ± 0.02	10.30 ± 0.04	16.87 ± 0.09	24.10 ± 0.04
9	3.02 ± 0.06	10.56 ± 0.09	16.75 ± 0.12	26.38 ± 0.21	1.44 ± 0.05	10.29 ± 0.04	16.69 ± 0.05	24.04 ± 0.20
12	3.17 ± 0.04	10.59 ± 0.10	16.84 ± 0.18	26.14 ± 0.05	1.47 ± 0.04	10.41 ± 0.11	16.87 ± 0.14	24.11 ± 0.06
18	2.92 ± 0.03	10.73 ± 0.14	16.82 ± 0.07	26.18 ± 0.16	1.44 ± 0.03	10.29 ± 0.07	16.51 ± 0.09	24.09 ± 0.17
$\bar{Y}$	3	10.62	16.91	26.07	1.46	10.41	16.71	24.09
$\bar{X}$	8	8	8	8	8	8	8	8
*b* _1_	−0.008	−0.001	−0.006	0.011	−0.001	−0.015	−0.004	−0.001
*b* _0_	3.062	10.631	16.958	25.984	1.472	10.537	16.747	24.09
*S*	0.249	0.158	0.216	0.287	0.043	0.134	0.177	0.032
*s*(*b*_1_)	0.017	0.011	0.015	0.02	0.003	0.009	0.012	0.002
*u* _lts_	0.309	0.196	0.268	0.356	0.053	0.166	0.22	0.04

^a^ AAC of the four rice reference materials was detected at the wavelengths of 620 and 720 nm, and the result are presented as means ± SD (n = 3) of g/100 g on dry matter bases. b_0_: intercept; b_1_: slope; S: standard deviation at each point on the regression line; s(b_1_): standard deviation of the slope; u_lts_: uncertainty of long-term stability; $\bar{X}$: mean value of the time; $\bar{Y}$: mean value of AAC.

**Table 3 molecules-27-04647-t003:** Analyses of short-term stability of AAC of the four rice reference materials ^a^.

Time (Days)	620 nm				720 nm			
	RM01	RM02	RM03	RM04	RM01	RM02	RM03	RM04
**0–4 °C**								
0	2.95 ± 0.06	10.96 ± 0.19	17.26 ± 0.19	26.50 ± 0.15	1.42 ± 0.04	10.26 ± 0.15	16.53 ± 0.05	23.97 ± 0.14
3	3.30 ± 0.08	10.74 ± 0.21	17.37 ± 0.23	25.73 ± 0.2	1.59 ± 0.03	10.24 ± 0.15	16.59 ± 0.22	24.10 ± 0.22
7	3.35 ± 0.11	10.77 ± 0.13	17.44 ± 0.15	26.12 ± 0.05	1.57 ± 0.05	10.24 ± 0.07	16.62 ± 0.12	24.10 ± 0.06
14	2.86 ± 0.09	10.69 ± 0.08	16.98 ± 0.08	26.05 ± 0.16	1.55 ± 0.07	10.14 ± 0.06	16.79 ± 0.15	24.01 ± 0.11
$\bar{Y}$	3.11	10.79	17.26	26.1	1.53	10.22	16.63	24.05
$\bar{X}$	6	6	6	6	6	6	6	6
*b* _1_	−0.013	−0.016	−0.022	−0.015	0.006	−0.008	0.006	0.001
*b* _0_	3.189	10.887	17.392	26.195	1.496	10.271	16.556	24.041
*S*	0.29	0.087	0.186	0.37	0.084	0.022	0.03	0.079
*s*(*b*_1_)	0.028	0.008	0.018	0.035	0.008	0.002	0.003	0.008
*u* _sts_	0.387	0.117	0.248	0.493	0.112	0.029	0.041	0.105
**19–21 °C**								
0	2.88 ± 0.01	10.79 ± 0.21	17.61 ± 0.18	26.21 ± 0.20	1.38 ± 0.02	10.25 ± 0.08	16.49 ± 0.03	24.06 ± 0.01
3	3.09 ± 0.06	10.77 ± 0.16	17.00 ± 0.25	25.80 ± 0.15	1.34 ± 0.00	10.22 ± 0.15	16.43 ± 0.15	23.89 ± 0.15
7	3.38 ± 0.05	10.75 ± 0.05	17.44 ± 0.12	26.12 ± 0.21	1.52 ± 0.03	10.18 ± 0.06	16.50 ± 0.02	23.86 ± 0.09
14	2.93 ± 0.05	10.94 ± 0.11	17.05 ± 0.10	25.91 ± 0.12	1.51 ± 0.03	10.63 ± 0.11	16.51 ± 0.09	24.05 ± 0.14
$\bar{Y}$	3.07	10.81	17.27	26.01	1.44	10.32	16.48	23.96
$\bar{X}$	6	6	6	6	6	6	6	6
*b* _1_	0.002	0.011	−0.025	−0.011	0.012	0.028	0.003	0.002
*b* _0_	3.053	10.745	17.425	26.08	1.367	10.153	16.464	23.951
S	0.277	0.069	0.311	0.214	0.069	0.152	0.035	0.126
*s*(*b*_1_)	0.026	0.007	0.03	0.02	0.007	0.014	0.003	0.012
*u* _sts_	0.37	0.093	0.415	0.285	0.092	0.203	0.046	0.168
**39–41 °C**								
0	2.85 ± 0.02	10.82 ± 0.17	17.63 ± 0.19	26.33 ± 0.25	1.71 ± 0.01	10.47 ± 0.15	16.54 ± 0.15	24.34 ± 0.19
3	3.00 ± 0.05	10.74 ± 0.12	17.18 ± 0.14	26.01 ± 0.16	1.55 ± 0.03	10.65 ± 0.09	16.87 ± 0.11	24.42 ± 0.14
7	3.38 ± 0.04	10.68 ± 0.06	17.60 ± 0.09	26.52 ± 0.10	1.39 ± 0.05	10.27 ± 0.05	16.50 ± 0.09	24.01 ± 0.06
14	3.00 ± 0.01	11.11 ± 0.15	17.66 ± 0.21	26.47 ± 0.11	1.66 ± 0.00	10.19 ± 0.12	16.51 ± 0.20	24.00 ± 0.09
$\bar{Y}$	3.06	10.84	17.52	26.33	1.58	10.39	16.61	24.19
$\bar{X}$	6	6	6	6	6	6	6	6
*b* _1_	0.011	0.022	0.014	0.021	−0.002	−0.026	−0.011	−0.03
*b* _0_	2.987	10.705	17.434	26.21	1.587	10.552	16.674	24.372
*S*	0.264	0.165	0.256	0.238	1.26	3.248	4.083	4.937
*s*(*b*_1_)	0.025	0.016	0.024	0.023	0.017	0.015	0.019	0.014
*u* _sts_	0.352	0.22	0.341	0.317	0.232	0.206	0.27	0.202

^a^ AACs of the four rice reference materials were detected at the wavelengths of 620 and 720 nm, and the results are presented as means ± SD (*n* = 3) of g/100 g on dry matter bases. b_0_, b_1_, s(b_1_), $\bar{X}$ and $\bar{Y}$: shown in Table 3; *u*_sts_: uncertainty of short-term stability.

**Table 4 molecules-27-04647-t004:** Analysis of AAC of the four rice reference materials from ten laboratories ^a^.

	620 nm				720 nm			
	RM01	RM02	RM03	RM04	RM01	RM02	RM03	RM04
L01	3.02 ± 0.22	10.56 ± 0.07	16.91 ± 0.28	26.38 ± 0.07	1.37 ± 0.12	10.18 ± 0.12	16.69 ± 0.08	23.83 ± 0.09
L02	3.25 ± 0.04	10.77 ± 0.06	17.27 ± 0.06	26.17 ± 0.06	1.68 ± 0.05	10.53 ± 0.06	16.73 ± 0.06	23.93 ± 0.06
L03	3.36 ± 0.04	10.88 ± 0.10	17.48 ± 0.11	26.55 ± 0.14	1.49 ± 0.05	10.69 ± 0.09	17.20 ± 0.12	24.63 ± 0.12
L04	2.89 ± 0.03	10.63 ± 0.17	16.59 ± 0.17	25.07 ± 0.10	1.62 ± 0.03	10.18 ± 0.14	16.37 ± 0.29	23.90 ± 0.14
L05	2.56 ± 0.06	10.32 ± 0.02	16.51 ± 0.11	25.21 ± 0.15	1.38 ± 0.12	10.35 ± 0.15	16.66 ± 0.08	23.94 ± 0.06
L06	3.37 ± 0.07	10.15 ± 0.08	16.88 ± 0.05	25.99 ± 0.08	1.57 ±0.02	10.23 ± 0.12	16.89 ± 0.10	23.96 ± 0.06
L07	2.65 ± 0.04	11.53 ± 0.04	17.61 ± 0.04	25.73 ± 0.04	1.53 ± 0.05	10.64 ± 0.05	16.28 ± 0.05	24.78 ± 0.00
L08	2.46 ± 0.41	10.26 ± 0.27	17.31 ± 0.11	26.03 ± 0.33	1.10 ± 0.05	10.26 ± 0.09	17.01 ± 0.08	24.51 ± 0.08
L09	3.19 ± 0.04	10.82 ± 0.04	17.32 ± 0.05	26.99 ± 0.08	1.72 ± 0.05	10.63 ± 0.09	16.70 ± 0.05	24.86 ± 0.08
L10	2.87 ± 0.21	10.84 ± 0.08	17.94 ± 0.05	26.77 ± 0.21	1.20 ± 0.06	10.68 ± 0.05	17.67 ± 0.09	24.96 ± 0.05
AVG	2.96	10.68	17.18	26.09	1.46	10.44	16.82	24.33
SD	0.33	0.399	0.454	0.627	0.203	0.216	0.406	0.457
CV	0.111	0.037	0.026	0.024	0.138	0.021	0.024	0.019
*u* _char_	0.104	0.126	0.144	0.198	0.064	0.068	0.128	0.145

^a^ AAC of the four rice reference materials was detected at the wavelengths of 620 and 720 nm, and the results are presented as means ± SD of g/100 g on dry matter bases (randomly 3 bottles with triplicate detection for each laboratory). AVG: average content; CV: coefficient of variation; SD: standard deviation; *u*_char_: uncertainty of inter-laboratory validation.

**Table 5 molecules-27-04647-t005:** AAC and the expanded uncertainties of the four rice reference materials ^a^.

	620 nm		720 nm	
	AAC	*U* (*k* = 2, %)	AAC	*U* (*k* = 2, %)
RM01	2.96	1.01	1.46	0.49
RM02	10.68	0.66	10.44	0.56
RM03	17.18	1.04	16.82	0.75
RM04	26.09	1.29	24.33	0.52

^a^ AAC of the four rice reference materials was detected at the wavelengths of 620 and 720 nm, and the results are presented as g/100 g on dry matter bases. AAC: apparent amylose content, *U*: expanded uncertainty.

**Table 6 molecules-27-04647-t006:** The application of the reference materials in detecting AAC of rice samples ^a^.

	620 nm				720 nm			
	AS	RM	AD	*r*	AS	RM	AD	*r*
SC	y = 0.0145x + 0.0824	y = 0.0106x + 0.0695	-	-	y = 0.0132x + 0.0244	y = 0.0072x + 0.0173	-	-
R	1	0.999	-	-	1	0.998	-	-
S01	1.84 ± 0.00 ^A^	1.63 ± 0.10 ^A^	0.21	0.77	1.21 ± 0.00 ^B^	0.99 ± 0.16 ^B^	0.22	0.74
S02	2.74 ± 0.20 ^A^	2.63 ± 0.15 ^A^	0.11	0.82	1.90 ± 0.39 ^B^	1.56 ± 0.11 ^B^	0.34	0.77
S03	7.97 ± 0.13 ^AB^	8.39 ± 0.29 ^A^	0.41	1.11	6.83 ± 0.10 ^C^	7.55 ± 0.21 ^B^	0.71	1.06
S04	9.34 ± 0.33 ^A^	9.25 ± 0.15 ^A^	0.09	1.17	8.08 ± 0.29 ^B^	8.04 ± 0.05 ^B^	0.05	1.1
S05	10.47 ± 0.20 ^A^	10.87 ± 0.00 ^A^	0.4	1.24	9.13 ± 0.20 ^C^	10.05 ± 0.11 ^B^	0.92	1.18
S06	14.01 ± 0.27 ^A^	14.18 ± 0.00 ^A^	0.17	1.42	13.71 ± 0.39 ^A^	14.06 ± 0.00 ^A^	0.35	1.41
S07	15.47 ± 0.07 ^B^	15.90 ± 0.10 ^A^	0.43	1.5	14.68 ± 0.00 ^C^	15.54 ± 0.16 ^B^	0.86	1.47
S08	16.51 ± 0.20 ^A^	16.21 ± 0.24 ^A^	0.3	1.54	15.72 ± 0.69 ^A^	15.46 ± 0.27 ^A^	0.26	1.5
S09	17.64 ± 0.33 ^A^	17.25 ± 0.05 ^AB^	0.39	1.59	16.49 ± 0.59 ^B^	16.33 ± 0.11 ^B^	0.15	1.54
S10	19.25 ± 0.07 ^A^	19.66 ± 0.15 ^A^	0.42	1.7	17.74 ± 0.20 ^C^	18.57 ± 0.16 ^B^	0.83	1.63
S11	25.00 ± 0.07 ^A^	24.11 ± 0.20 ^C^	0.89	1.96	24.47 ± 0.10 ^B^	22.77 ± 0.11 ^D^	1.7	1.91

^a^ AAC of rice samples was detected at the wavelengths of 620 and 720 nm, and the results are presented as g/100 g on dry matter bases. The values in each row with different letters are significantly different (*p* < 0.05). AD: absolutely difference between the two different methods; AS: the method with defatting procedure and with potato amylose and waxy rice amylopectin as standards; SC: standard curve; R: correlation coefficient, *r*: repeatability; RM: the method without defatting and with the four rice reference materials as standards, S01–S11: 11 rice varieties.

## Data Availability

The data supporting the findings of this study are available within the article and the Appendix A, or from the corresponding authors upon request.

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
