# Peer review of "Development of Certified Reference Materials for the Determination of Apparent Amylose Content in Rice"

_molecules, 2022, doi:10.3390/molecules27144647_

Round 1
Reviewer 1 Report
The objective of this study was to establish a set of certified reference materials for rice AAC determination at two absorption wavelengths. Generally, the manuscript is readable and the obtained results presented and discussed properly. However, before publication the manuscript needs some corrections and supplementations.
Detailed recommendation:
Abstract: please add more data/results to the abstract.
Key words: add: starch
Introduction: Provide more information about effect of amylose content on properties of rice
Statistical analysis: what was statistical analysis? In how many repetitions was made research?
Add SD value to the tables with mean values.
Author Response
The objective of this study was to establish a set of certified reference materials for rice AAC determination at two absorption wavelengths. Generally, the manuscript is readable and the obtained results presented and discussed properly. However, before publication the manuscript needs some corrections and supplementations.
Reply: Thanks for your comments! We checked the whole manuscript carefully, answered all of the raised questions, and revised the manuscript point by point which were shown in red font.
Detailed recommendation:
Abstract: please add more data/results to the abstract.
Reply: We added sample quantity of homogeneity test, the temperature and the time of the short- and long-term stability test, and the results of homogeneity and stable test. The detection range of rice samples by using the undefatted rice reference materials was also exhibited. The details are as follows:
“Homogeneity test (n = 20) was judged by F-test based on the mean squares of among and within bottles, and short- and long-term stability monitoring was performed by T-test to check if there was significant degradation at the delivery temperature of under 40 °C (14 days) and at 0-4 °C storage condition (18 months), respectively.”
“It found that the four reference materials were homogenous and stable”
“It confirmed that the undefatted rice reference materials could get the satisfactory results to test the rice materials with the AAC ranging from 1 to 25 g/100 g.”
Key words: add: starch
Reply: The key word of “starch” was added in Keywords.
Introduction: Provide more information about effect of amylose content on properties of rice
Reply: Thanks! AAC, one of the most important parameters, can affect the quality of cooked rice. In the Introduction, we reviewed the effect of rice AAC on the texture of cooked rice and the related molecular mechanisms. The effects of rice AAC on pasting properties, gelatinization, solubility, and the formation of resistant starch were also pointed out. We added the information about the effect of AAC on the processed products, which was shown as follows: AAC could affect rice properties, and rice with different AAC could be processed into different kinds of rice products. For example, high AAC rice was preferred for processing of noodle or bread, whereas waxy and/or low AAC rice was preferred for processing of biscuits or crackers [9].
Statistical analysis: what was statistical analysis? In how many repetitions was made research?
Reply: Thanks for pointing out this. We added the “2.9 statistical analysis” as follows:
2.9. Statistical analysis
The AAC of the 11 rice sample were presented on a dry matter basis as means ± standard deviation of triplicate determinations. Data analysis were performed with SAS version 8 software (SAS Institute, Cary, NC, U.S.A.). Differences of the same sample by two different test methods at the 620 and 720 nm were found by using ANOVA, followed by Duncan multiple comparison tests. Statistical significance was defined at a level of P < 0.05.
Add SD value to the tables with mean values.
Reply: Thanks! All the data were presented as means ± standard deviations.
Reviewer 2 Report
The authors report the development of certified reference materials for the determination of apparent amylose content in rice. Since certified reference materials are very important for quantitative analysis, this work is thus very interesting. After carefully read, I think the work can be published in the present form.
Author Response
Thank you very much for the kind-hearted comments.
Reviewer 3 Report
Manuscript molecules-1808897 deals with the developemnt of certified materials for the determination of apparent amylose content in rice. Rice is a basic food worldwide and the studies dealing with its quality control and typification are of importance. The authors run different inter-laboratories validation to conclude that the data obtained from the undeftted rice samples are more robust. The article is of interest to the readership of Molecules. It is an original study. Limited studies are available in the literature that compare the results of analyzed food samples with those of reference materials.The article is well written and data have been discussed properly.However, it would be of great value if the authors can give a predictive model based on the regression analysis results or run a classification analysis, for the apparent amylose content. Other comments the authors will find in the attached pdf.
Based on the significanve of the study, I suggest a minor revision prior to further consideration for publication.

Author Response
Thanks very much for the comments and the valuable suggestion. The construction of rice AAC predictive model needs more data, and it would be studied in further.
We checked the whole manuscript carefully, and revised the manuscript point by point according to the attached PDF file. The corrections were shown in red font. Thank you!
Reviewer 4 Report
Manuscript ID: molecules-1808897
Decision Letter of Article: Development of certified reference materials for the determina-2 tion of apparent amylose content in rice
This paper showed an interesting study related to the development of the certified method for determination of amylose in rice. It is an interesting work that must be complemented with references performed recently.
The manuscript needs to complement the references with some recent work involving the amylose determination using NIR and chemometric tools. I suggest the following papers:
Sampaio et al (Food Chemistry) 2018
Sampaio et al (2019) (European F. R. Technology)…among others.
This work is very interesting and must be improved and corrected according to the reviewer's suggestions. For those reasons, I suggest to Editor that this paper must be accepted with minor revision.
Author Response
Thanks for the comments. The papers of Sampaio et al (Food Chemistry) 2018 and Sampaio et al (2019) (European F. R. Technology) were cited in the text and listed in the reference list.